# A-IDE : Agent-Integrated Denoising Experts

**Uihyun Cho**[*1]                                                          CUIHYUN12@YONSEI.AC.KR

**Namhun Kim**[*1]                                                          KSOUTH0413@YONSEI.AC.KR

[1] *Yonsei University, Seoul, South Korea*

## Abstract

Recent advances in deep learning-based denoising methods have improved the quality of Low-Dose CT (LDCT) images. However, due to anatomical variability and limited data availability, a single model often struggles to generalize effectively across multiple anatomical regions. To address this limitation, we propose the Agent-Integrated Denoising Experts (A-IDE) framework. A-IDE integrates three region-specialized RED-CNN models under the control of a decision-making large language model (LLM) agent. This agent analyzes anatomical priors extracted from BiomedCLIP and dynamically routes incoming LDCT scans to the most suitable specialized model. We highlight three major advantages. First, A-IDE shows robust performance in heterogeneous and data-scarce environments. Second, the framework reduces risk of overfitting by distributing tasks among multiple experts. Finally, the fully automated agent-driven routing eliminates the need for manual intervention. Experimental results in the Mayo-2016 dataset confirm that A-IDE achieves superior performance in RMSE, PSNR, and SSIM compared to a single unified denoiser.

**Keywords:** Low-Dose CT, Image Denoising, LLM, Agents

## 1. Introduction

LDCT significantly reduces patient radiation at the cost of increased noise and artifacts, which can obscure diagnostic details and adversely affect clinical decision making (Zhang et al., 2024a). Although deep learning-based methods like RED-CNN (Chen et al., 2017), GANs (You et al., 2019), Transformers (Zhang et al., 2021), and Diffusion models (Zhao et al., 2023) have shown notable success in denoising LDCT images, they typically rely on single and generalized architecture. However, anatomical regions differ greatly in terms of Hounsfield Unit (HU) intensity distributions, scanner protocols, and artifact characteristics. As a result, not a single method consistently performs well across all regions (Deng and Campbell, 2025). Furthermore, data scarcity and privacy constraints often lead models to overfit on a limited range of anatomies or characteristics, particularly in underrepresented or imbalanced categories (Won et al., 2021). These regional variations and limited data highlight the need for context-aware, region-specific denoising frameworks (Chen et al., 2023) (Yang et al., 2025).

To overcome these limitations, we propose A-IDE, which employs a decision-making LLM agent to evaluate the anatomical context of each incoming LDCT image and orchestrate the optimal denoising experts. First, the input CT image is processed by BiomedCLIP (Zhang et al., 2024b) to generate an anatomical prior distribution over anatomical structures. This distribution, along with textual descriptions of three pretrained RED-CNN

---

[*] Contributed equally

denoising models—each specialized for a distinct anatomical region—is passed to a LLM agent (gpt-4o) (OpenAI, 2024). Guided by prompts detailing each model's anatomical focus, the agent dynamically selects the most appropriate expert model for the given input. The selected model then reconstructs the denoised image patches and reports quantitative metrics. Our experimental results show that A-IDE achieves superior performance in terms of PSNR, SSIM, and RMSE compared to baseline approach.

## 2. Methods

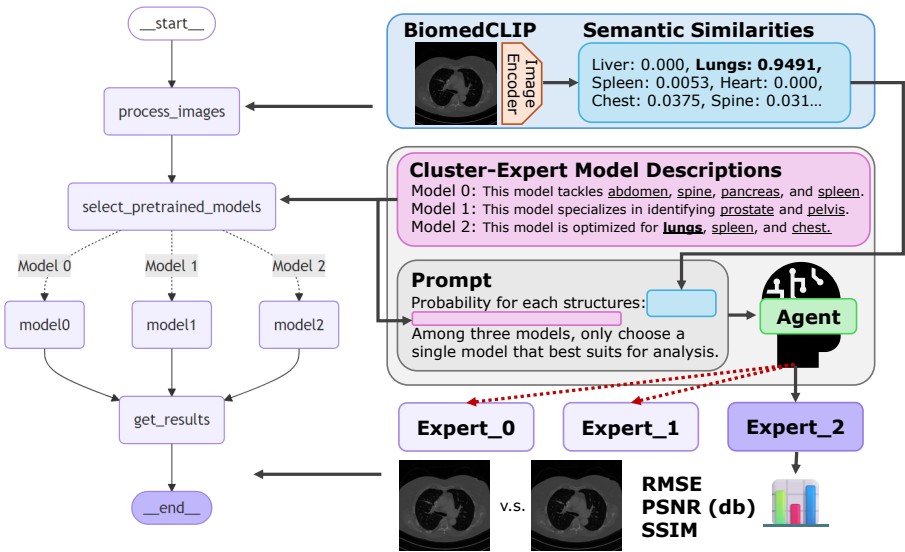

Figure 1: A-IDE Architecture.

We use the 2016 Low-Dose X-ray CT Grand Challenge dataset (Mayo-2016) (McCollough et al., 2017), which includes paired LDCT and NDCT images from 10 anonymized patient scans. Each $512{\times}512$ image is normalized and divided into non-overlapping $55{\times}55$ patches to yield 81 patches per image. To simulate anatomy-specific denoising, we cluster the dataset into three groups based on semantic embeddings extracted from BiomedCLIP. These high-dimensional vectors are reduced to two dimensions using Principal Component Analysis (PCA) and clustered using K-means (k=3). We characterize each cluster by averaging the anatomical structure probabilities over a randomly selected set of 100 images, thereby capturing the dominant anatomical features within each group.

We train a baseline model and three cluster-specific expert models using the RED-CNN architecture. The models are optimized using the Adam optimizer with an initial learning rate of 1e-5, a mean squared error (MSE) loss, and a learning rate scheduler that decays the rate to with a minimum threshold 1e-10. We then introduce an LLM-driven agent that autonomously selects the most suitable expert model based on the anatomical content of each CT image. Each image is first processed by BiomedCLIP to extract semantic embeddings and generate a probability distribution over twenty anatomical structures. This distribution, along with concise descriptions of the three specialized RED-CNN models, is

passed as a prompt to a GPT-4o agent. Guided by this contextual information, the agent selects the optimal model to perform patch-wise denoising.

## 3. Results and Discussion

| Methods | RMSE | PSNR | SSIM |
|---|---|---|---|
| Baseline | $0.097_{\pm 0.00164}$ | $43.06_{\pm 1.73}$ | $0.9557_{\pm 0.0125}$ |
| Expert 0 | $0.097_{\pm 0.00245}$ | $42.15_{\pm 2.28}$ | $0.9483_{\pm 0.0200}$ |
| Expert 1 | $\mathbf{0.086}_{\pm 0.00165}$ | $43.33_{\pm 1.82}$ | $0.9576_{\pm 0.0107}$ |
| Expert 2 | $0.107_{\pm 0.00196}$ | $40.89_{\pm 1.39}$ | $0.9435_{\pm 0.0147}$ |
| A-IDE | $0.094_{\pm 0.00169}$ | $\mathbf{43.42}_{\pm 1.72}$ | $\mathbf{0.9583}_{\pm 0.0110}$ |

Table 1: Evaluation metrics (mean $\pm$ standard deviation) for the baseline, cluster-specific expert models, and the proposed A-IDE framework. A-IDE consistently outperforms the baseline in RMSE, PSNR, and SSIM, and achieves the highest PSNR and SSIM scores among all models.

Table 1 presents the quantitative performance of all models in terms of RMSE, PSNR, and SSIM. The single, generalized baseline model demonstrates competitive performance across all metrics. However, the cluster-specific expert models, especially Expert 1, exhibit superior performance due to training tailored to specific anatomical regions. The proposed A-IDE framework effectively leverages the strengths of these individual anatomical experts to deliver superior overall performance. Notably, A-IDE outperforms all individual expert models in both PSNR and SSIM, demonstrating the effectiveness of the proposed framework's adaptive integration strategy in harnessing the complementary strengths of multiple specialized denoisers to achieve more accurate and reliable reconstruction—particularly in the presence of complex, overlapping, or anatomically ambiguous regions. As a result, A-IDE achieves enhanced visual fidelity and structural preservation compared to both the baseline and individual expert models.

## 4. Conclusion

Our results demonstrate the effectiveness of employing an intelligent agent to dynamically route inputs to the most appropriate expert model. By harnessing the complementary strengths of multiple specialized RED-CNN experts, the agent-driven A-IDE framework achieves a balanced trade-off between low reconstruction error, high signal-to-noise ratio, and accurate structural preservation. As a result, A-IDE delivers robust performance across diverse imaging scenarios, positioning it as a promising solution for enhancing LDCT reconstruction in real-world clinical applications.

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
