# OpenReview forum: "A-IDE : Agent-Integrated Denoising Experts"
_MIDL.io/2025/Short_Papers — MIDL 2025 - Short Papers_

### Official Review · Reviewer_B8DX · 2025-04-29

**Rating:** 4
**Confidence:** 4

**Summary:**

The paper introduced an agent-integrated low-dose CT denoising framework with three region-specialized RED-CNN models and a decision-making LLM agent.

**Strengths:**

* Interesting paper with clear clinical motivation
* The proposed A-IDE method achieved a balanced trade-off between low reconstruction error, high SNR, and structural preservation
* Experimental results suggest improvement over specialized denoising RED-CNN models

**Weaknesses:**

* The performance of the proposed A-IDE is not convincing. A-IDE has a higher RMSE score than Expert 1 and is comparable to the baseline model. Although PSNR and SSIM scores increased, the improvement is marginal or no improvement at all.
* It would be interesting to experiment with more single-organ experts as well as with different choices of backbones.

---

### Decision · Program_Chairs · 2025-05-01

Accept